# Estimation of Head Accelerations in Crashes Using Neural Networks and Sensors Embedded in the Protective Helmet

**DOI:** 10.3390/s22155592

**Published:** 2022-07-26

**Authors:** Andrea Bracali, Niccolò Baldanzini

**Affiliations:** Department of Industrial Engineering, University of Florence, 50139 Firenze, Italy; niccolo.baldanzini@unifi.it

**Keywords:** traumatic brain injuries (TBIs), linear acceleration, rotational acceleration, safety, helmet sensors, neural networks

## Abstract

Traumatic Brain Injuries (TBIs) are one of the most frequent and severe outcomes of a Powered Two-Wheeler (PTW) crash. Early diagnosis and treatment can greatly reduce permanent consequences. Despite the fact that devices to track head kinematics have been developed for sports applications, they all have limitations, which hamper their use in everyday road applications. In this study, a new technical solution based on accelerometers integrated in a motorcycle helmet is presented, and the related methodology to estimate linear and rotational acceleration of the head with deep Artificial Neural Networks (dANNs) is developed. A finite element model of helmet coupled with a Hybrid III head model was used to generate data needed for the neural network training. Input data to the dANN model were time signals of (virtual) accelerometers placed on the inner surface of the helmet shell, while the output data were the components of linear and rotational head accelerations. The network was capable of estimating, with good accuracy, time patterns of the acceleration components in all impact conditions that require medical treatment. The correlation between the reference and estimated values was high for all parameters and for both linear and rotational acceleration, with coefficients of determination ($R^{2}$) ranging from 0.91 to 0.97.

## 1. Introduction

Despite attempts to minimize the incidence and severity of head injuries with improved protective equipment, closed-head impacts represent the highest percentage of Traumatic Brain Injuries (TBIs) diagnosed each year among the civil population in the United States (US) [1]. In 2017, 61,000 TBI-related deaths occurred in the United States, and motor vehicle crash was the second most relevant category after suicide [2]. In addition, TBIs, regardless of severity level, can lead to difficulties in performing daily activities, such as gait impairment [3]. Worldwide, Vulnerable Road Users (VRUs) account for more than half of all global deaths in road crashes (the events related to two- and three-wheeled vehicles represent 26% of all deaths) [4]. Therefore, models capable of properly estimating TBI are needed to perform real-time estimation of the injuries and thus to improve protective devices.

Currently, TBI risk assessment is made using criteria coupling a biomechanical metric and an injury risk function. There are two types of biomechanical metrics: based on kinematic parameters of the head or on the brain tissue deformation during the impact. Most of the existing injury criteria are based on the head kinematics since measurements, either on a dummy or a volunteer, are easier than measuring brain tissue response. An overview of these metrics was provided by Gabler et al. [5]. The latest findings on the key role played by rotational acceleration on brain injuries led the United Nations Economic Commission for Europe to revise the ECE 22.05 helmet homologation standard [6]. The new regulation, ECE 22.06, took effect on January 2021 and introduced new tests for homologation also based on the rotational acceleration [7]. Several studies [8,9,10,11,12,13,14] were conducted to improve the knowledge of TBI and to develop a method or an injury criterion to estimate injuries based on kinematic parameters. The possibility of estimating the linear and rotational accelerations of the head during a crash becomes key to predicting TBIs, but the estimation process is extremely difficult in real-world conditions (e.g., impacts between football players, motorcyclists’ road crashes, skiers’ falls). Since a helmet is the most common solution to mitigate TBIs, several technical solutions used the helmet as part of the measuring system. Systems incorporating microelectromechanical system (MEMS) inertial sensors into helmets were developed and employed, with the Head Impact Telemetry System (HITS) [15] being one of the earliest and most widely used [11,16,17,18,19,20,21]. The HIT system is composed of six single-axis accelerometers oriented normal to the skull, and it is specifically designed to measure head accelerations by elastically coupling the accelerometers in contact with the head, isolating them from the helmet shell. The linear acceleration is estimated with an optimization method, while the rotational acceleration is computed assuming a pivot point located about 10 cm below the head Centre of Gravity (CoG). A development of HIT with a new sensor layout resulted in the 6DOF (Degrees of Freedom) HIT measurement device [22], providing both the linear and the rotational accelerations and iteratively solving the optimization problem [12,20,23].

The most recent helmeted device was the gForce Tracker (GFT) [24], characterized by a triaxial accelerometer and a triaxial gyroscope embedded inside a casing attached to the helmet. This technology provides the maximum values of the resultant linear acceleration and rotational velocity obtained from a power fit regression. Other devices require a rigid connection to the head such as mount-guards, earplugs and bands. These devices provide more accurate data at the expense of comfort and user-friendliness, and they may cause an acceptability problem to end users such as motorcyclists or bikers.

Although the indirect identification of the head kinematics by using helmet dynamics was extensively deepened in previous research, all measurement devices fitted to the helmet and capable of estimating kinematic parameters as a function of time require sensors to be in contact with the head to overcome the difficulties due to the relative movement between head and helmet. The helmet rotation in relation to the head is primarily affected by the coupling of different head and helmet sizes as well as by a combination of padding compression, chin strap tension, and friction between the helmet and the head. This dependency produces different rotational velocities and rotational acceleration sustained by the helmet and head during an impact. Additionally, the presence of the foam component between the helmet outer shell and the head significantly reduces its linear acceleration compared to the helmet one. Manoogian et al. [25] demonstrated that the helmet peak linear acceleration is approximately 10 times the head peak acceleration. More recently, Joodaki et al. [26] found that the peak linear acceleration of the helmet was 2/5 times greater than the head one, while the helmet peak angular velocity was greater or smaller than the head one according to the impact conditions; in some tests, the helmet rotated more than 30 deg relative to the head.

The technical solution investigated in this paper is based on a novel sensing system characterized by a new layout of the accelerometers, which are placed on the inner side of the helmet shell. In this configuration, the signal processing for the determination of the head kinematics cannot rely on simplifying assumptions, as typical of previous studies. The high number of parameters influencing head kinematics increases the complexity level and suggests the use of deep learning techniques for the estimation task.

At present, there are a limited number of applications of deep Artificial Neural Networks (dANNs) for identifying the impact load history of structures [27,28,29]. dANNs are computing models used for information processing that only need data for supervised learning. They are often used to identify and model a complex functional relationship or pattern between input and output data with a black-box approach. ANNs can detect complex non-linear relationships between variables through their training phase. Nowadays, several software offer packages to develop ANNs in an easy and friendly way with multiple training algorithms [30]. However, ANNs also have some disadvantages: performances achieved are highly reliant on the quantity and quality of data given as input and output to train the networks; collecting the necessary amount of data to train the network can be costly and time-consuming [31]. Staszewski [28] and Ghajari [27] used an dANN model to identify the impact load acting on a composite plate; specifically, in Ghajari [27], the effects of signal features, network architectures and sensor placements on the performance of the dANN model were analyzed. Most recently, Zhou [29] proposed a novel impact load identification method of non-linear structures by using deep Recurrent Neural Networks (RNNs), verifying this method in three non-linear cases: damped Duffing oscillator, non-linear three-degree-of-freedom system and non-linear composite plate. Finally, deep learning neural networks techniques were recently implemented to recognize human activities from wearable sensors [32,33]. To date, there is no application for the identification of a body (e.g., head) acceleration with sensors embedded in a second body (e.g., helmet), which exhibits a relative movement to the first one.

This paper contributes to the methodological development of a device for the real-time estimation of TBIs. Specifically, it proposes a novel method to estimate, as time functions, the components of head linear and rotational accelerations from signals provided by twelve single-axis accelerometers embedded in the outer shell of a motorcycle helmet. The sensors are organized in orthogonally oriented pairs at six different locations, with their sensing axis tangential to the helmet, as described in [34]. The system’s non-linearity at the helmet–head interface is modeled within the estimation process using deep learning ANNs. They were trained using data collected from finite element simulations, which is a time-saving and cost-effective technique compared to carrying out the same number of experimental tests.

## 2. Materials and Methods

Head impact scenarios were defined considering previous research on head impacts. A wide and representative set of impact conditions was defined and reproduced with a finite element model to generate the necessary datasets. The Artificial Neural Network architecture was defined to optimize the estimation of head kinematics. The procedure implemented to develop and validate the technology is described at the end of this section.

### 2.1. Head Impact Scenarios

A head impact scenario can be defined by two different variables: relative position between the helmeted head and the ground and the impact speed vector. Considering a reference scenario (Figure 1a) where the helmeted head was oriented such that the transverse plane of the head was parallel to the ground, the relative position in a generic scenario was defined by sequentially rotating the ground around the X-axis and Y-axis, while the helmeted head position was unchanged. Different ranges were considered for these two parameters: the angle β around the X-axis was varied between $-100^{\circ }$ and $100^{\circ }$, but the angle γ around the Y-axis was varied between $-125^{\circ }$ and $115^{\circ }$.

The impact speed vector was defined through its tangential and normal components to the ground and the orientation of the tangential component in the ground plane. Vehicle impact speed varies between 20 and 88 km/h in urban accidents [35]. Different studies reproducing typical oblique impacts in motorcycle crashes through experimental impact tests [36] or FE analyses [27,37] observed that the angular kinematics of the head remained quite constant at high tangential speeds. Cernicchi et al. [38] simulated motorcycle head impacts using two different $V_{n}$ magnitudes: 2.20 and 5.66 m/s and a tangential velocity ranging from 0 to 60 m/s. They observed that at constant $V_{n}$, the angular acceleration peak did not vary above a specific $V_{t}$ threshold ($V_{t}^{*}$). However, as $V_{n}$ increased, the value of $V_{t}^{*}$ seemed also to increase (for $V_{n}$ = 2.20 m/s, $V_{t}^{*}$ = 4.21 m/s, while for $V_{n}$ = 5.66 m/s, $V_{t}^{*}$ = 8.57 m/s). Based on previous research, the magnitude of the impact speed ranged between 8 and 78 km/h to reproduce real-world head impacts in urban crashes. This magnitude was obtained by combining the normal speed component $V_{n}$, which ranged between 2 and 12 m/s, and the tangential speed component $V_{t}$, which varied between 3 and 18 m/s. A range of $360^{\circ }$ was considered for the orientation of the tangential component in the ground plane (angle θ).

The latter five parameters (rotational angles around X-axis and Y-axis, normal and tangential impact speed components and tangential speed orientation) were combined using the Latin Hypercube Sampling (LHS) method [39] to define three different datasets of, respectively, 2000, 200 and 300 simulations. The purpose of the three datasets will be clarified later in Section 3. The impact points on the helmet outer shell for the training dataset of 2000 simulations are shown in Figure 1b.

### 2.2. Finite Element Head and Helmet Model

Head impact scenarios were simulated using a Hybrid III (HIII) Head finite element model (https://biocorellc.com/finite-element-models/) (accessed on 21 July 2022) distributed by the Biomechanics and Research, LLC (Biocore) [40], which was coupled with an AGV X3000 full-face helmet provided by the Dainese company.

The original HIII model included both the head and neck, but only the head model was used in this study, in accordance with the ECE 22.06 standard [7]. The HIII head was previously validated by the University of Virginia Center for Applied Biomechanics [40], simulating the NHTSA Head Drop Certification Test. The present head includes three main parts: skin rubber layer, rigid skull and head mount. Both the head skin and mount used hexahedral solid elements, but the quadrilateral shell elements were used to mesh the rigid skull.

The helmet model consists of an outer shell, an energy-absorbing liner, a chin pad and a chin strap. Further information about the helmet finite element model used in this paper can be found in [34]. This model was updated, including the accelerometers of the measurement device. Further information about the procedure adopted to select the accelerometer locations can be found in [34].

### 2.3. Neural Networks

Recurrent neural networks [41] are one of the most known types of Artificial Neural Networks, capable of processing sequential data or time series data. The ability to use feedback loops, commonly described as "memory", ensures the output is influenced by both previous and current inputs. The process of carrying memory forward is described mathematically:(1)$$h_{t}=f_{h}\left(U*x_{t}+V*h_{t-1}\right)$$
(2)$$o_{t}=f_{o}\left(W*h_{t}\right)$$
where $x_{t}$ is the input at time step t, $h_{t}$ stores the values of the hidden units at time step t and $o_{t}$ is the output at time step t. $f_{h}$ and $f_{o}$ are the hidden and output unit activation functions. They define how the weighted sum of the input is transformed into an output; commonly used activation functions are logistic sigmoid, rectified linear (ReLU) and Hyperbolic Tangent (Tanh). The weight matrices *U*, *W* and *V* are determined with supervised training of the RNN, but it requires a huge amount of data (Figure 2a). A relevant problem in RNNs concerns long-term memory: input information $\left(x_{t^{\prime }}\right)$ persists for a short time, but it cannot be kept for a long period of time due to the vanishing gradient problem.

The long-term dependency was improved with the Long Short-Term Memory (LSTM) structure, introduced by Hochreitener and Schmidhurber [42] for the first time in 1997. They replaced the typical recurrent unit of RNNs with a more complex gated recurrent unit. The LSTM used in this study is a standard LSTM, and it is shown in Figure 2b. The LSTM forward propagation process is described mathematically:(3)$$f_{t}=\sigma \left(W_{f}*\left[h_{t-1},x_{t}\right]+b_{f}\right)$$
(4)$$i_{t}=\sigma \left(W_{i}*\left[h_{t-1},x_{t}\right]+b_{i}\right)$$
(5)$$\tilde{C_{t}}=tanh\left(W_{c}*\left[h_{t-1},x_{t}\right]+b_{c}\right)$$
(6)$$C_{t}=f_{t}*C_{t-1}+i_{t}*\tilde{C_{t}}$$
(7)$$o_{t}=\sigma \left(W_{o}*\left[h_{t-1},x_{t}\right]+b_{o}\right)$$
(8)$$h_{t}=o_{t}*tanh\left(C_{t}\right)$$
where *x* and *h* are the input and the output, *W* the weight matrices and *b* the biases; σ and tanh are the logistic sigmoid and Tanh activation functions. The LSTM cell can decide whether to discard or keep the past information using the variable “Cell State” (*C*). With a value of C equal to 1, the information is completely kept, while a 0 means that the information is discarded. The signals of the accelerometers embedded in the helmet are the inputs to the LSTM network, while the outputs are the components of the linear or rotational acceleration of the head. Two separate but identical networks were used for the linear and rotational components to improve the estimation results.

A simple Neural Network with a single LSTM layer did not provide good performances in the impact force history identification, as stated by Zhou [29]. Assuming that the same conclusion can be considered for the estimation of impact acceleration, the deep neural network architecture used by Zhou [29] was the starting point for this study. Some improvements were identified to maximize the performance for the specific problem, and the final architecture is shown in Figure 3. It consists of a Bidirectional Long Short-Term Memory (BLSTM) layer, two LSTM layers and two Fully Connected (FC) layers. Basically, a BLSTM layer consists of a forward LSTM layer and a backward LSTM layer. This provides the opportunity to consider both past and future responses of sequential inputs. BLSTM and the two LSTM layers have 200 cells per layer, the first FC layer has 200 hidden units, but the last FC layer has a number of hidden units equal to the number of outputs. The over-fitting was prevented using the dropout operation to each non-recurrent connection, as shown in the dashed lines in Figure 3. From here onwards, this neural network architecture will be referred to as deep Artificial Neural Network (dANN).

## 3. Identification Steps

STEP 1—Development of simulation datasets. Three datasets of simulations reproducing real-world head impacts in urban crashes were implemented with the procedure explained in Section 2.1. The first dataset consisted of 2000 simulations, and the second and the third ones consisted of, respectively, 200 and 300 simulations. The first dataset was used for training the network, while the second and the third datasets were used, respectively, for validation and testing purposes.

STEP 2—Neural Networks design. Two deep neural networks with the same structure described in Section 2.3 were implemented. The networks had the same inputs, i.e., accelerations from the accelerometers embedded in the helmet’t outer shell, but the outputs differed: the three head linear acceleration components were used as outputs of the network *A* and the three head rotational acceleration components were the outputs of the network *B*. From here onwards, the neural networks estimating linear and rotational accelerations will be referred to as *A* and *B*, respectively.

STEP 3—Data preparation. Accelerations of the helmet outer shell were filtered using a SAE108 filter, but both the linear and rotational head accelerations were filtered with a CFC1000 filter as suggested by ISO 6487:2002 standard [43].

STEP 4—Model training. Networks *A* and *B* were fed with the inputs and outputs described in STEP 2. Weights and biases were optimized using a back-propagation through time (BPTT) [44] algorithm. The Root Mean Square propagation (RMSprop) [45] was selected as the optimizer. RMSprop is a Gradient Descent-based Learning Algorithm, which adapts, at each iteration, the learning rate of each parameter individually using a subset of the training data. A different subset, called a mini-batch, is used at each iteration. A mini-batch size equal to 16 was chosen to implement this process. The initial learning rate was 0.001, and the maximum epochs used for the training were 1000. After each training epoch, the Root-Mean-Squared-Error (RMSE) on the validation data was monitored. If the RMSE did not decrease after 30 consecutive epochs, the training process was stopped. Every 45 epochs, the learning rate was reduced by a factor of two.

STEP 5—Model performance assessment. Helmet accelerations from the test dataset were used as input for both the NNs *A* and *B*, and the estimated linear and rotational head acceleration components were compared to the target ones. Three parameters were introduced to assess the model performance: peak error, Head Injury Criterion (HIC) [46] and Rotational Injury Criterion (RIC) [12]. The assessment parameters were applied to both the reference head accelerations (extracted from the simulations) and the head accelerations estimated with the dANN. For each of them, the Pearson coefficient $R^{2}$ was calculated.

HIC [46] is the most commonly used metric for evaluating head and brain injury risk; it is currently required by the UN/ECE 22.06 standard [7] used in helmet regulation.
(9)$$HIC=max\left\{\left(t_{2}-t_{1}\right){\left[\frac{1}{t_{2}-t_{1}}\int_{t_{1}}^{t_{2}}\left|a(t)dt\right|\right]}^{2,5}\right\}$$
where *a* is the magnitude of the resultant head linear acceleration, and $t_{1}$ and $t_{2}$ are, respectively, the initial and final integral times over which HIC is calculated ($t_{1}$ and $t_{2}$ are selected to maximize HIC).

RIC was proposed by Kimpara et al. [12]; it was formulated similarly to HIC by replacing the linear acceleration term with angular acceleration.
(10)$$RIC=max\left\{\left(t_{2}-t_{1}\right){\left[\frac{1}{t_{2}-t_{1}}\int_{t_{1}}^{t_{2}}\left|\alpha (t)dt\right|\right]}^{2,5}\right\}$$
where α is the magnitude of the resultant head linear acceleration. For $t_{1}$ and $t_{2}$, the same considerations made for the HIC are applicable.

In this paper, no consideration about the correlation between the HIC/RIC and head injuries will be analyzed. These criteria were selected exclusively as parameters to compare the estimated and target curves.

## 4. Results

### 4.1. Training Dataset

Training, validation and testing datasets were generated as described in Section 2.1 and the five parameters used to define the impact simulations were combined using the LHS. Initially the 2000 values of each parameter were sampled uniformly within their ranges. With this solution, most of the impact points were in the top outer shell area, but the left, right, back and chin areas had a poor spatial sampling. Therefore, the parameters that contributed to spreading the impact points over the outer shell surface (i.e., α and β) were modified, as shown in Figure 4a for β. Angle distributions were modified to have lower frequencies between $-20^{\circ }$ and $20^{\circ }$ than between $-100^{\circ }$ and $-60^{\circ }$ or $60^{\circ }$ and $100^{\circ }$. The new distribution of the parameters created a uniform spatial sampling over the entire outer shell surface of the helmet, as shown in Figure 1.

Figure 4b,c show, respectively, the distribution of the impact speed and the angle between the impact speed direction and the ground. Most of the impact simulations are characterized by an impact speed magnitude within the range of 8–16 m/s and an inclination angle within the range of 16–45${}^{\circ }$.

### 4.2. Head Linear Acceleration

The deep Artificial Neural Networks used to estimate the head linear acceleration components were trained, as described in STEP 4 of Section 3. LSTM and BLSTM layers and the first fully-connected layer had 200 hidden units. Dropout probability was set equal to 0.5 and learning rate equal to 0.001. Two examples of the estimated linear acceleration are shown in Figure 5: the plot in the rightmost column shows the resultant accelerations obtained by combining the three components plotted in previous columns.

The dANN reproduces the acceleration component’s shape and the peak values in both cases well (Figure 5). Peak values and HIC were calculated on the entire testing dataset (300 simulations). Scatter plots in Figure 6 show the comparison between peak values of the target and estimated acceleration components. Peaks of the Z-component had the best overall performance with $R^{2}=0.972$. Parameters calculated on the resultant acceleration, i.e., peak value and HIC, had similar $R^{2}$ high values, confirming the strong learning ability of dANNs (Figure 7).

### 4.3. Head Rotational Acceleration

The dANN used to estimate the head rotational accelerations had the same architecture as the dANN described in Section 4.2. Figure 8 shows the rotational accelerations for the same cases reported in Figure 5. Rotational acceleration components of shape and peak values were estimated with a good approximation (Figure 9). Rotational peak values around the Y-axis had the best performances, with $R^{2}=0.908$. $R^{2}$ coefficients for the peak values around the X and Z axes were, respectively, 0.800 and 0.783.

Peak values of the resultant acceleration and RIC have an $R^{2}$ correlation coefficient, respectively, of 0.771 and 0.687. (Figure 10). For both parameters, the regression line and the symmetry axis largely diverge for high values of acceleration.

## 5. Discussions

The indirect head kinematics estimation, through sensors embedded in protective devices and far from the head, is the only way to enable the real-time evaluation of TBIs without reducing the acceptability of the protective devices by the end users. This study presents the development and assessment of a new methodology, based on deep neural networks applied to a sensor system, to estimate the head kinematics with sensors attached to the inner surface of a helmet’s outer shell.

Different designs for the neural network used to estimate linear and rotational accelerations were tested. The analyzed architectures differed in the number of BLSTM and LSTM layers. Table 1 reports the designs of the networks and their performances for the rotational acceleration. $R^{2}$ for peak values and RIC were used to compare the performances of the different deep neural networks investigated. DANN0 has the architecture and hyperparameters suggested by Zhou [29]; it was trained using the Adam optimizer, i.e., the same used by Zhou in his study. Network performances were not acceptable since values of $R^{2}$ ranged between 0.168 and 0.278 for the estimation of peak values. A set of changes (network architecture integrated with an FC layer, learning rate decreased to 0.001, hidden units decreased to 100, and the Adam optimizer replaced with the RMSprop one) led to great improvements. Progressive increase in Hidden Unit values (100, 200 and 300; DANN1-DANN3) generated a slight improvement in the performances against an increase in training time. For instance, using 300 hidden units instead of 200 did not produce considerable improvement (as described in the next paragraphs, results are mainly affected by the lack of data for high acceleration values); therefore, 200 hidden units were used to analyze the learning rate influence. The worst results were obtained by decreasing or increasing the learning rate, as shown, respectively, by model DANN4 and DANN5. Other more complex architectures were investigated (models DANN6-DANN9), but none of them considerably improved the performances to justify the adoption of a more complex architecture against an increase in training time. BLSTM, 2LSTM and 2FC architecture were the most suitable for head kinematics estimation, and it was selected as the preferred one. DANN2 (Table 1) was used both to present the results shown in Section 4 and for the considerations listed below.

The overall correlation between the reference and estimated head accelerations was higher for the linear accelerations than the rotational ones (i.e., $R^{2}=0.946$ vs. $R^{2}=0.772$ for the resultant). Specifically, rotational acceleration estimation worsens as the absolute value of the acceleration peak increases. As shown in Figure 10, the regression line and the symmetry axis diverge for high values of acceleration. This did not happen for the parameters of the resultant linear acceleration (Figure 7), as the regression line and the symmetry axis remained almost parallel, even for high values of acceleration. The increase in the identification error was expected since rotational acceleration is strongly influenced by friction between head and helmet and the system behavior is more complex to model.

Looking at the comparison between the target and estimated peak values of the resultant rotational acceleration (Figure 10a), the network underestimates the target values starting from a target peak of 40 krad/s${}^{2}$. This is confirmed by a smaller slope of the regression line compared to the symmetry axis. The network performance is affected by the small number of impacts with high rotational acceleration in the datasets (143 out of 2000 in the training dataset, 28 out of 300 in the testing dataset with a peak of rotational acceleration exceeding 40 krad/s${}^{2}$). This result is a consequence of the procedure used to generate the databases: the position of the helmet with respect to the ground and the impact speed is defined, and the accelerations of the head–helmet system are a consequence of these parameters. Nonetheless, the network performances need to be properly framed within the physiological limits of the human body and take into account the overall objective of the research (i.e., early detection and treatment of TBIs). From this context, acceleration values, which cause immediate and permanent brain damage, should be clearly excluded.

Brain injury tolerance based on head rotational acceleration was investigated in numerous studies. Pike [47] proposed a peak angular head acceleration of 9 krad/s${}^{2}$ associated with a 10% risk of Mild Traumatic Brain Injury (MTBI) based on 27000 head impacts recorded from American football players at the collegiate level. Zhang et al. [48] proposed a maximum resultant rotational acceleration peak of 7.9 krad/s${}^{2}$ for an 80% probability of sustaining an MTBI. Rotational acceleration peak was also connected to specific TBI such as concussion and DAI. The first type of injury was analyzed by Ommaya et al. [8], which suggested an angular acceleration tolerance of 1.8 krad/s${}^{2}$ for a 50% probability of concussion. Rowson et al. [20] proposed a tolerance value of 6383 krad/s${}^{2}$ for the same probability of concussion. Finally, Margulies and Thibault [10] proposed an angular acceleration of 16 krad/s${}^{2}$ as tolerance to moderate to severe DAI for the human head subjected to a lateral motion. These results support the exclusion of impacts with peak rotational acceleration above 40 krad/s${}^{2}$ from the assessment of the proposed method, as this threshold is more than double the highest value cited in [10]. With the redefined dataset, the estimation of peak values is improved, and $R^{2}$ exceeded 0.9 for each component. The new scatter plots for peak values of the acceleration components, the resultant acceleration and RIC are shown in Figure 11 and Figure 12. Considering Figure 7 and Figure 12, a slight systematic underestimation of the accelerations is still evident when the absolute peak value increases. Future exploitation of this method in a real-time system implemented into a helmet should include a proper correction to adjust for these errors.

A considerable amount of data is usually needed to obtain acceptable results from a trained deep neural network. Zhou et al. [29], in the identification of the impact load history acting on a composite plate, have used 10,000 signals to train their dANN, compared to 2000 acceleration signals used in this study. All simulations were run on a 72-cpu cluster and each simulation took approximately 20 min, for 28 days of total computational time. As well as the performances of the dANNs, calculation time is really important for a future industrial application of this method. Results proved that the indirect identification of head kinematics history through dANNs, fed with data from accelerometers embedded in the helmet, can be successfully accomplished using a training dataset of limited size. The use of the LHS method to combine the factors described in Section 2.1 and create a well-distributed dataset of simulations for the training process was a relevant contribution to the methodology, as it avoided data overfitting and selection bias.

## 6. Conclusions

In this paper, a novel method to identify the head linear and rotational acceleration time signals from acceleration data, acquired by single-axis sensors embedded in the outer shell of a helmet and processed with deep learning techniques, is presented. The results support the following major findings:Neural Networks: LSTM Neural Networks are capable of reproducing the underlying non-linear behavior of the model. A specific network was defined to solve the problem, and the related hyperparameters were determined. This result enables real-time prediction of head kinematics and paves the road to the application of specific metrics for TBI estimation.Dataset:-The training, validation and testing datasets can be obtained from a virtual environment using state-of-the-art tools (e.g., Finite Elements). The approach used for the generation of datasets greatly reduces time requirements and costs compared to experimental tests;-An adequate size for the datasets was determined, which may be used for guidance in further applications. The networks can be trained with a limited amount of data because of the well-distributed dataset generated using the LHS method to combine the parameters defining our case study;-Both the use of virtual testing tools and the application of the LHS method to generate the simulation inputs facilitate the industrial application of the methodology.Prediction: The results proved the high accuracy of the trained networks, as a high correlation coefficient was obtained for all the parameters used in the assessment stage.

To the best knowledge of the authors, this is the first application of neural networks for the estimation of a body (head) acceleration with sensors embedded in a second body (helmet), considering that the two bodies are not fully connected. The results proved the feasibility of the proposed methodology. The next steps will focus on the experimental validation of the findings highlighted in this study.

## Figures and Tables

**Figure 1 sensors-22-05592-f001:**
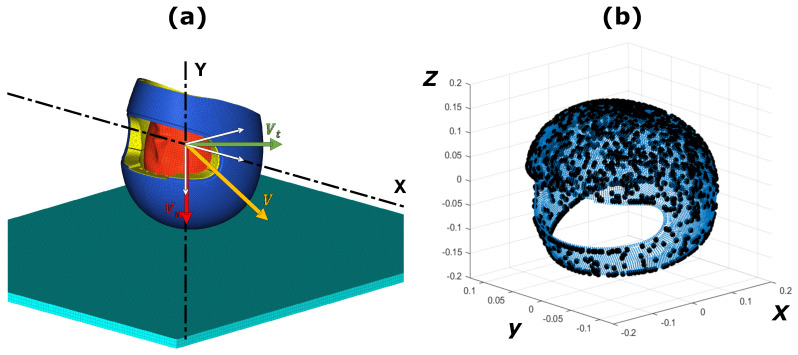
(**a**) Start of the first head-impact scenario; (**b**) scatter plot of the shell mesh nodes (blue points) and the impact points (black points) for the entire training dataset.

**Figure 2 sensors-22-05592-f002:**
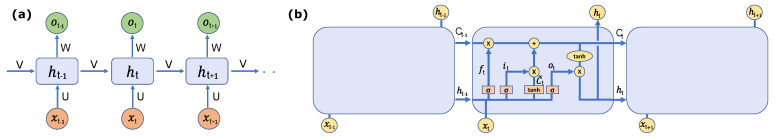
(**a**) Recurrent Neural Network; (**b**) Long Short Term Memory (LSTM) cell.

**Figure 3 sensors-22-05592-f003:**
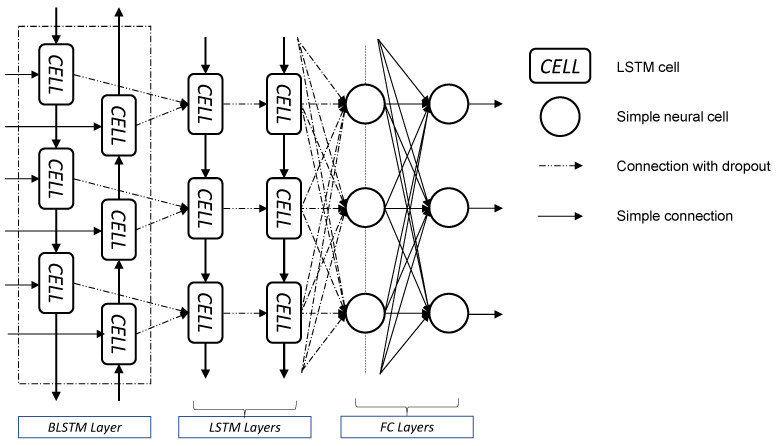
Deep Artificial Neural Network (dANN) architecture.

**Figure 4 sensors-22-05592-f004:**
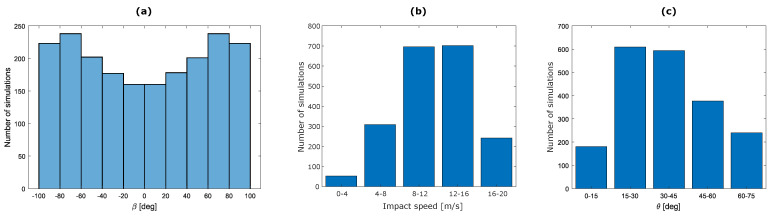
Distribution of (**a**) β angle values, (**b**) the impact speed magnitude, (**c**) the angle between the impact speed direction and the ground for the training dataset simulations.

**Figure 5 sensors-22-05592-f005:**
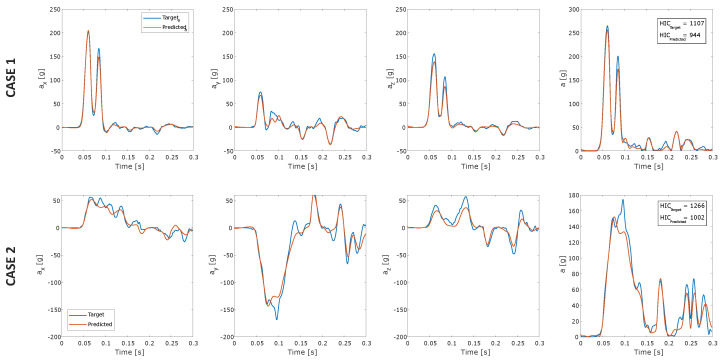
Linear acceleration prediction using the deep neural network model.

**Figure 6 sensors-22-05592-f006:**
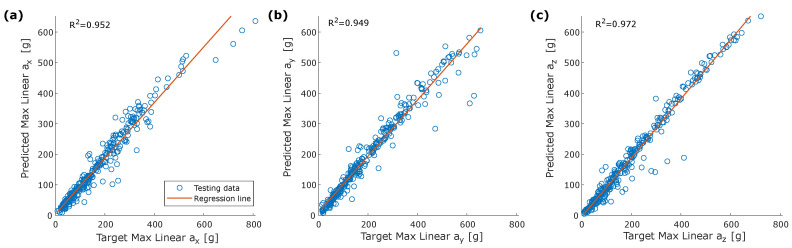
Correlation between target and predicted peak linear acceleration: (**a**) $a_{x}$; (**b**) $a_{y}$; (**c**) $a_{z}$.

**Figure 7 sensors-22-05592-f007:**
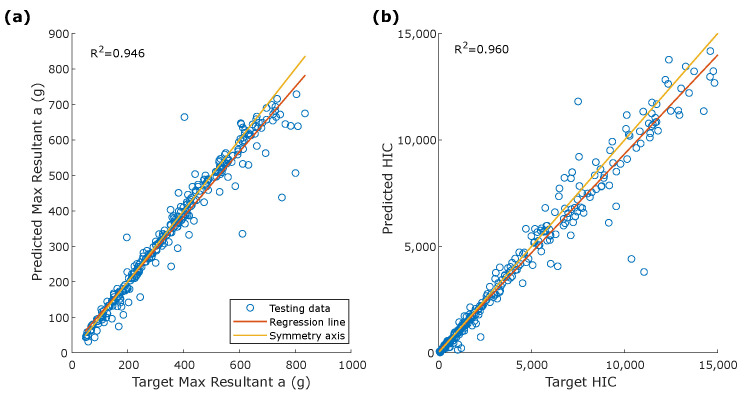
Correlation between (**a**) target and predicted peak for the resultant linear acceleration and (**b**) target and predicted HIC.

**Figure 8 sensors-22-05592-f008:**
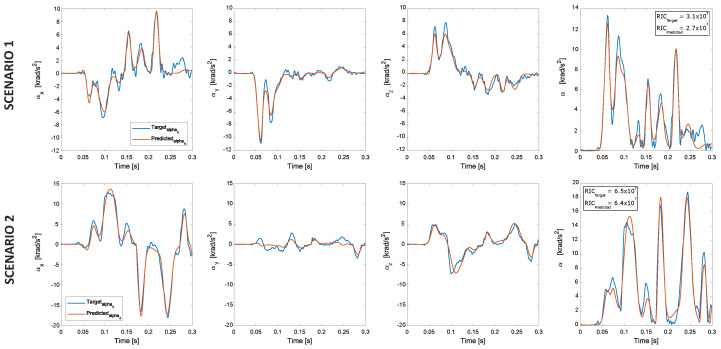
Rotational acceleration prediction using the deep neural network model.

**Figure 9 sensors-22-05592-f009:**
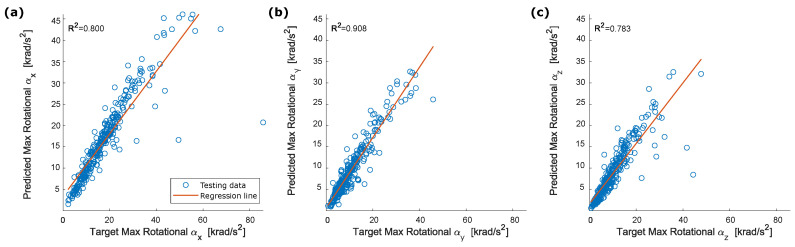
Correlation between target and predicted peak rotational acceleration: (**a**) $\alpha_{x}$; (**b**) $\alpha_{y}$; (**c**) $\alpha_{z}$.

**Figure 10 sensors-22-05592-f010:**
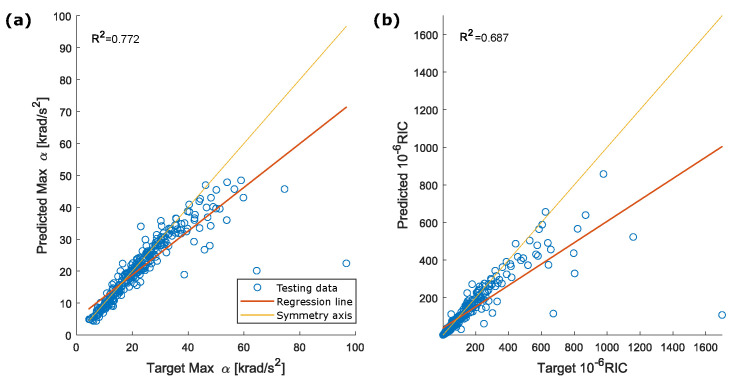
Correlation between (**a**) target and predicted peak for the resultant rotational acceleration and (**b**) target and predicted RIC.

**Figure 11 sensors-22-05592-f011:**
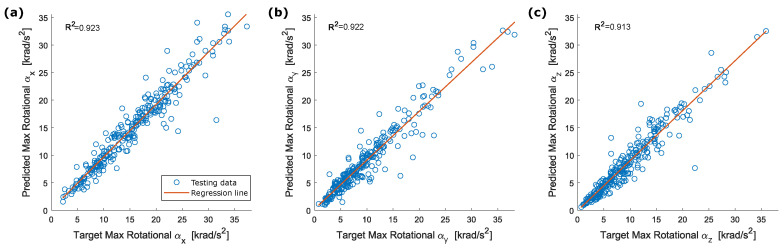
Correlation between target and predicted peak rotational acceleration for the reduced testing dataset (exclusion of impacts with peak rotational acceleration above 40 krad/s${}^{2}$): (**a**) $\alpha_{x}$; (**b**) $\alpha_{y}$; (**c**) $\alpha_{z}$.

**Figure 12 sensors-22-05592-f012:**
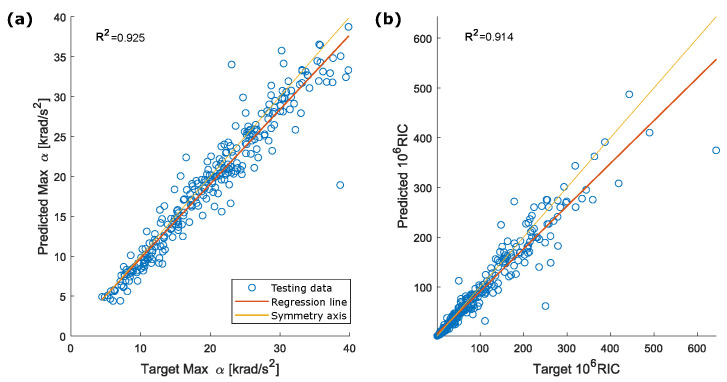
Correlation between (**a**) target and predicted peak for the resultant rotational acceleration and (**b**) target and predicted RIC for the reduced testing dataset (exclusion of impacts with peak rotational acceleration above 40 krad/s${}^{2}$).

**Table 1 sensors-22-05592-t001:** Effect of learning rate, hidden units and architecture in the peak values and RIC estimation.

	Architecture	Learning Rate	Hidden Units	$R^{2}$ Peak Value				$R^{2}$ RIC
				$\alpha_{x}$	$\alpha_{y}$	$\alpha_{z}$	**Resultant**	
DANN0	BLSTM and 2LSTM [29]	0.005	128	0.248	0.278	0.168	0.192	0.404
DANN1	BLSTM and 2LSTM and 2FC	0.001	100	0.776	0.882	0.795	0.755	0.672
DANN2	BLSTM and 2LSTM and 2FC	0.001	200	0.800	0.908	0.783	0.772	0.687
DANN3	BLSTM and 2LSTM and 2FC	0.001	300	0.802	0.927	0.827	0.783	0.681
DANN4	BLSTM and 2LSTM and 2FC	0.01	200	0.775	0.849	0.759	0.746	0.679
DANN5	BLSTM and 2LSTM and 2FC	0.0001	200	0.689	0.016	0.231	0.529	0.514
DANN6	BLSTM and 3LSTM and 2FC	0.001	200	0.817	0.901	0.834	0.782	0.710
DANN7	2BLSTM and 3LSTM and 2FC	0.001	200	0.791	0.917	0.834	0.791	0.705
DANN8	3BLSTM and 2LSTM and 2FC	0.001	200	0.804	0.890	0.765	0.772	0.650
DANN9	4LSTM and 2FC	0.001	200	0.782	0.900	0.780	0.772	0.643

## Data Availability

The data that support the findings of this study are available from the authors upon reasonable request.

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
