# Peer review of "Estimation of Head Accelerations in Crashes Using Neural Networks and Sensors Embedded in the Protective Helmet"

_sensors, 2022, doi:10.3390/s22155592_

Round 1
Reviewer 1 Report
In my opinion the paper is adequate for publication.
The work seems to be original and it is clearly written.
The proposed solution is well supported by the theory.
The only weak point is the bibliography that could be improved by adding some papers
In the same topic that can be found also in mdpi.
Reviewer 2 Report
The article under the title “Estimation of head accelerations in crashes using neural networks and sensors embedded in the protective helmet” discusses a unique method for identifying and analyzing the head’s linear and rotational acceleration time histories from the data collected by sensors installed inside and outside the shell of a helmet via deep learning techniques. The article is well-written and can be accepted for publication after the following revisions:
1. In the introduction section, there is a need to highlight the advantages and disadvantages of neural networks. For your reference, you may look at the article (DOI: 10.3390/ma14010163).
2. The article presents a novelty that must be highlighted in the last paragraph of the introductions section such as “the dataset has been collected using finite element model, which is computationally efficient and cost-effective technique compared to the experimental analyses.”
3. How it can be assured that the results predicted by the finite element model are accurate and close to the real-time analyses? Here, we require the real-time analysis comparison as well.
4. Can you provide the justification for why the physics-informed neural network was not applied for such a study as it fits very well to the current scenario?
5. The conclusion section needs significant improvement. It is important to highlight the major findings of this study using bullet points.
